# Factors influencing the acceptability of alcohol drinking for a patient with colorectal cancer

**Camille Auriol©\*, Nicole Cantisano©, Patrick Raynal**

Centre d'Etudes et de Recherches en Psychopathologie et Psychologie de la Santé, Université de Toulouse-Jean Jaurès, Toulouse, France

\* camille.auriol1@univ-tlse2.fr

## Abstract

### Introduction

Colorectal cancer is the second deadliest cancer worldwide. One of the risk factors for the development of this type of cancer is alcohol consumption. Patients with colorectal cancer may be stigmatized regarding their cancer and regarding drinking behaviors they may exhibit. This study aimed to analyze community persons' and health professionals' acceptability judgments regarding alcohol drinkers having colorectal cancer.

### Method

This study relies on an experimental method enabling the identification of variables involved in one's judgment, based on the exhaustive combination of factors yielding several scenarios rated by participants. Scenarios implemented factors possibly influencing participants' perception of a woman character having colorectal cancer. Factors included her drinking habits, post-diagnosis drinking behavior and type of diagnosis/prognosis. The participants were community persons (N' = 132) or health professionals (N" = 126). Data were analyzed using a within-subject factorial ANOVA.

### Results

In both samples, the "Post-diagnosis behavior" factor had large effect sizes, with drinking cessation being more acceptable than other drinking behaviors. Another factor, "Drinking habits", had significant influences on participants judgments, as higher drinking was considered less acceptable. A third factor, "Diagnosis" (polyps, early- or late-stage cancer), was taken into account by participants when it interacted with "Drinking habits" and "Post-diagnosis behavior". Indeed, participants considered most acceptable to continue drinking in the case of late-stage cancer, especially in the health professional sample where the acceptability of continuing drinking was almost doubled when the character had advanced- rather than early-cancer.

**Data Availability Statement:** All data set and logfiles are available from the Zenodo database (accession number https://doi.org/10.5281/zenodo.7900049).

**Funding:** CA was supported by a fellowship program from Institut National du Cancer (INCa_15747 SPADOC20-02). https://www.e-cancer.fr/"). The funders had no role in study design, data collection and analysis, decision to publish, or preparation of the manuscript.

**Competing interests:** The authors have declared that no competing interests exist.

## Conclusion

The lesser the drinking behavior, the better the acceptability. However, advanced cancer stage attenuated the poor acceptability of drinking in both samples, as participants' attitudes were more permissive when the patient had advanced cancer.

## Introduction

Globally, colorectal cancer represents 1.8 million of new cancer cases every year [1]. This cancer is the third most commonly diagnosed, accounting for 11% of all cancer cases, and the second most deadly [1]. It has been shown that alcohol is a risk factor for the development of colorectal cancer [2–7] and that approximately 17% of colorectal cancers may be attributable to people's alcohol consumption [8].

Patients diagnosed with colorectal cancer, in addition to experiencing physical pain, may undergo negative psychological consequences, such as depression and anxiety [9–12]. These negative consequences may be even greater for patients who feel stigmatized and judged by others [13–16]. Stigma can be defined as a discrediting attribute marking someone as different, which can lead to a spoiled identity, i.e., the sense of being inferior, "defective", and socially undesirable [17,18]. Stigma is engendered by others through the actions or judgment of the person concerned, resulting in an impact on the stigmatized person's dignity [19]. Stigma is characterized by the exclusion, rejection, blame or devaluation that result from the experience, perception or reasonable anticipation of an unfavorable social judgment on a person [20]. An important component of stigma is social acceptability, which refers to the extent to which something is deemed acceptable or tolerable [21].

Stigma can severely alter colorectal cancer patients' quality of life, especially their social and emotional spheres [22]. For instance, the stigma attached to bowel cancer patients may be due to beliefs about cancer causality, as studies have shown that, in community samples, there are beliefs that this type of cancer is caused by alcohol consumption or an unhealthy diet. Another belief is that colorectal cancer may result from anal intercourse [23–26]. Due to these causal representations present among the general population, people living with colorectal cancer may be confronted with reactions of disgust from others, thus increasing the feeling of perceived stigmatization [24]. Furthermore, it was reported that negative judgments and perceived stigma concerning colorectal cancer could instigate a delay in diagnosis, as patients may fear being stigmatized [27–30].

Similarly to the stigma felt by colorectal cancer patients, people with high alcohol use have been shown to be highly stigmatized [31,32]. As in the case of colorectal cancer, the perceived stigma around alcohol use can result in depressive disorders [33] or reduce quality of life [34] in individuals who drink. As with colorectal cancer, different studies have shown that judgments and stigma about alcohol use constitute a barrier to seeking help and treatment for alcohol addiction [35–41].

Studies focusing on attitudes towards people who drink have shown that health professionals [42] and healthcare students [43] prefer not having to deal with patients with alcohol problems, or find such patients troublesome to manage [44].

Negative attitudes towards drinkers are also present among the general public, who, for instance, expresses more social distance in comparison to other disorders (*e.g.*, anorexia or obsessive-compulsive disorder) [45]. It has also been shown that the level of alcohol consumption influences public attitudes, given that a person diagnosed with an alcohol use disorder

would be more stigmatized than casual drinkers [46]. A gender effect was also shown in the stigma of drinkers, as attitudes appeared to be more negative when it comes to a woman consumer [47].

Moreover, it has been shown that patients were more stigmatized when having cancer considered to have been caused by preventable behaviors [48], which may be the case of colorectal patients drinking alcohol. The fact that people with alcohol addiction tend to be seen by others as responsible for their disorder supports this view [49]. Considering the negative influence of stigmatization of alcohol use and colorectal cancer highlighted in the aforementioned studies, the reduction of stigmatization seems to be an important issue in terms of healthcare for drinkers with colorectal cancer [50]. However, achieving this goal would first require to identify factors influencing stigmatizing attitudes around drinking among the general public and health professionals. As a first attempt to reach this objective, this study aimed at analyzing how people judge acceptable, or not, the behaviors of a drinker having colorectal cancer. This study's methodology relies on Functional Measurement (FM) which is based in the Functional Theory of Cognition (FTC), a theory with attempts to apprehend how individuals process information when forming judgments [51,52]. The method is based on an exhaustive combination of factors presented as realistic scenarios to participants who are instructed to rate them on an acceptability scale. This study used scenarios depicting the drinking habits of a 60-year-old fictitious woman recently diagnosed with colorectal cancer. A female character was chosen since women drinkers are most stigmatized [47]. The scenarios were based on four "within subject factors" that potentially have an impact on the perception of a person who drinks alcoholic beverages and who is diagnosed with colorectal cancer. Two factors were directly linked to drinking behaviors, i.e., the drinking habits (e.g., one drink per day or several drinks per day) and the drinking behavior following the cancer diagnosis (e.g., to quit drinking, to think about quitting, or to continues drinking). The first factor, i.e., the drinking habits, was chosen considering that a former study focused on self-stigma showed there may be differences in stigmatization between individuals who only occasionally use alcohol and those with an alcohol use disorder [46]. The second factor, i.e., the drinking behavior following cancer diagnosis, was selected based on studies of stigmatizing attitudes regarding fictional patients with lung cancer, as studies showed that patients who smoke were judged more negatively than those who were non-smokers or who quit smoking at the time of diagnosis [53,54]. A third factor was the type of cancer diagnosis and prognosis, as cancer stage has been linked to different levels of psychological distress [55]. The fourth factor was physical activity or sedentary lifestyle, considering that physical exercise showed significant benefits for patients having cancer [56] and was identified as a factor decreasing the risk of developing colorectal cancer [2]. In addition, the choice of this factor was based on reports on stigmatizing attitudes that showed that a sedentary lifestyle was a behavior that participants identified as avoidable and as a risk factor for cancer [48,57]. The choice of these four factors was also justified by a recent report focused on acceptability judgments regarding the behaviors of a fictional smoker diagnosed with lung cancer. Indeed, this study showed that participants' judgments were influenced by four similar factors, including smoking habits, smoking behavior following cancer diagnosis, cancer stage and prognosis, and physical activity [58]. We thus assumed that an analogy exists between factors related to drinking in colorectal cancer and those related to smoking in lung cancer, which further supported the choice of the four factors listed above.

The aim of this study was thus to determine the importance of each factor in individuals' judgments regarding a fictitious character presenting both drinking habits and colorectal cancer. The study is based on two different samples: community persons and health professionals.

## Method

### Transparency and openness promotion

In this article, we report how we determined our sample size, all manipulations, and all measures that were included in the study. There was no data exclusion. All the data and code from this study are available and can be accessed at https://doi.org/10.5281/zenodo.7900049. Data were analyzed using IBM SPSS Statistics version 26.

### Participants

Community participants were recruited through advertisements on Instagram® and Facebook® lifestyle groups. Health professionals were recruited through advertisements on LinkedIn® or on Facebook® professional health caregiver groups. Participants who replied to the announcements were contacted via private messaging on these social networks. If they agreed to participate, they were included in the study by the first author (CA), a Ph.D. student and certified psychologist who conducted all experiments and data gathering. All participants were adults from all age groups and gender, living in France. Health professionals had different occupations (i.e., nurses, paramedical professionals, medical doctor, caregivers, nurse assistants, and health executives). Participants did not receive compensation for their participation and their answers were anonymous. The sample size was determined using G*Power [59]. The data were collected from October 2021 to May 2022.

### Material

The experimental material was composed of 36 scenarios describing the alcoholic beverage consumption habits of a 60-year-old fictitious woman recently diagnosed with colorectal cancer. The story also described her drinking behavior following diagnosis and her physical activity habits. The scenarios were constructed following standard procedures (FM; FTC), implying an orthogonal combination of the following four factors, [51,52]: 1- "Drinking habits" had two levels: (1) 1 drink/day or (2) 5–6 drinks/day; 2- "Type of diagnosis/prognosis" had three levels: (1) polyps that may not turn into cancer but require follow-up, (2) early-stage colorectal cancer with life expectancy greater than several years, or (3) advanced colorectal cancer with life expectancy no greater than several months; 3- "Post-diagnosis drinking behavior" had three levels: (1) quits drinking, (2) thinks about quitting, or (3) continues drinking; 4- "Physical activity" had two levels: (1) physical exercise or (2) sedentary lifestyle. The exhaustive combination of these factors (orthogonal crossing) resulted in the following factorial design: 2 x 3 x 3 x 2 = 36 scenarios, covering all possible combinations of factors and levels.

An example of a scenario is the following (translated from French language): "Elisabeth is a 60-year-old woman, working as an office employee. Elisabeth enjoys physical exercise and, during her free time, she attends a walking club twice a week. For the last 30 years, Elisabeth has been having a daily consumption of 5–6 drinks of alcoholic beverages. She tried several times to quit drinking, without success. For the past three weeks, she has been experiencing abdominal pain with diarrhea and bleeding. She thus undertook a medical evaluation. The medical results showed the presence of polyps. These lesions will require a follow-up, so that they do not turn into colorectal cancer. She thus makes an appointment for a follow-up consultation. A few weeks later, Elisabeth continues to drink. Likewise, she continues to spend her leisure time as she did before."

Each scenario was printed on a separate sheet of paper. Below each story, the following question was asked: "Taking into account all these elements, according to you, does Elisabeth behave in an acceptable manner?". Underneath the question, an 11-point (non numerical)

response scale was printed with the sole labels "Not at all acceptable" in the left end and "Totally acceptable" in the right end. The experiment was pilot-tested with 20 volunteers, including healthcare practitioners. Their opinions were asked regarding format, length, clarity and credibility of the scenarios, which allowed to improve the material and procedure.

## Procedure

The experiment took place in the university's premises or at the participant's workplace, depending on what was most convenient for the participant. The procedure consisted of two phases [51]. The first phase aimed at familiarizing the participants with the material. Nine scenarios randomly chosen were presented to the participants. They were then asked to read each scenario and place an X on the response scale to make their acceptability judgment. During this phase, participants were allowed to compare their ratings and change their responses until they were satisfied with their ratings. In the second phase (experimental phase), all 36 scenarios were given randomly to participants who were again instructed to read each scenario and place an X on the response scale. During this phase, they were no longer allowed to compare their responses nor ask questions to the experimenter. Participants worked individually, at their own pace.

Following completion of both tasks, participants were asked to answer questions collecting sociodemographic data (age, sex, marital status, education level and occupational category) and their own alcohol consumption using the brief AUDIT-C questionnaire [60].

## Statistical methods

Data were screened for missing values and outliers due to plausible invalid entries of participant's responses. Factorial analyses of variance (ANOVA) were conducted in order to analyze participants' responses to all 36 scenarios and according to the following within-subject design: Drinking habits (1 *vs.* 5–6 drinks/day) x Diagnosis (polyps *vs.* early-stage cancer *vs.* advanced cancer) x Post-diagnosis behavior (quit *vs.* think about quitting *vs.* continue drinking) x Physical activity (physical exercise *vs.* sedentary lifestyle). Tukey's test was used as post-hoc. The analysis of interactions between factors was performed using factorial ANOVA followed by pairwise comparison of means using paired t-tests. Differences were considered significant if $p < 0.05$.

## Results

### Participants' characteristics and descriptive statistics

Participants were assigned either to the health professional or the community sample, depending whether or not they worked as health professionals. The community sample was composed of 132 participants (Mean age = 35.45 years, SD = 15.60, range 18–85). The health professional sample contained 126 participants (Mean age = 38.37 years, SD = 12.29, range 22–72). In both samples, the majority of participants were female and lived in a couple (Table 1). In the health professional sample, a majority of participants had a drinking level considered as non-problematic (AUDIT-C total score below 4). In the community sample, participants with a drinking level considered as non-problematic represented nearly 47% of the whole sample. Participants with heavy drinking/high dependence represented < 3% of health professionals and < 8% of the community sample.

The mean acceptability judgment was 6.39 (SD = 1.94) in the community sample, and 7.10 (SD = 1.57) in health professionals: a *t* test (*t* = -3.22, $p < 0.05$) showed statistically significant differences between these two means. In the community sample, women's mean acceptability judgements were 6.49 (SD = 1.91) and men's were 6.22 (SD = 2.00), without any statistical differences between these means (*t* = 0.75, *p* = 0.45). In health professionals, women's and men's

**Table 1. Characteristics of the two samples.**

| | Community (N' = 132) | | | Health professionals (N" = 126) | |
|---|---|---|---|---|---|
| | *n* | % | | *n* | % |
| Sex | | | | | |
| Female | 82 | 62.12 | | 88 | 69.84 |
| Male | 50 | 37.88 | | 37 | 29.37 |
| Transgender | - | - | | 1 | 0.79 |
| Marital status | | | | | |
| Single | 58 | 43.94 | | 36 | 28.57 |
| As a couple | 74 | 56.06 | | 90 | 71.42 |
| Number of children | | | | | |
| None | 78 | 59.09 | | 60 | 47.61 |
| 1 or 2 | 40 | 30.30 | | 53 | 42.06 |
| 3 or more | 14 | 10.61 | | 13 | 10.31 |
| Education | | | | | |
| High school or less | 43 | 32.58 | | 18 | 14.29 |
| Undergraduate degree | 60 | 45.45 | | 33 | 26.19 |
| Graduate degree or above | 29 | 21.97 | | 75 | 59.52 |
| Occupational category | | | | | |
| Independent | 4 | 3.03 | Paramedical pro. & nurse | 88 | 69.84 |
| Employee, high | 16 | 12.12 | Medical doctor | 17 | 13.49 |
| Employee, intermediate | 17 | 12.88 | Caregiver & nurse assist. | 12 | 9.52 |
| Employee, low | 24 | 18.18 | Health executive | 9 | 7.14 |
| Retired | 10 | 7.58 | | | |
| Unemployed | 4 | 3.03 | | | |
| Student/Vocational training | 57 | 43.18 | | | |
| AUDIT-C total score | | | | | |
| 0–3 | 62 | 46.97 | | 75 | 59.52 |
| 4–7 | 60 | 45.45 | | 48 | 38.10 |
| 8–12 | 10 | 7.58 | | 3 | 2.38 |

mean acceptability judgements were 7.08 (SD = 1.57) and 7.18 (SD = 1.57), respectively, without any statistical differences between these means ($t$ = 0.32, $p$ = 0.75).

The effect of participants' educational level on acceptability judgments was tested in the community sample and the health professional sample. In the community sample, the acceptability judgments means for participants from the "High school or less", "Undergraduate degree", "Graduate and above" categories were 6.37 (SD = 2.11), 6.39 (SD = 1.94) and 6.41 (SD = 1.72), respectively. An ANOVA showed there was no difference between means (F[2,129] = 0.00; $p$ = 1). In the health professional sample, the acceptability judgments means for participants from the "High school or less", "Undergraduate degree", "Graduate and above" categories were 6.25 (SD = 0.31), 7.30 (SD = 1.41) and 7.21 (SD = 1.64), respectively. An ANOVA and post hoc test showed that participants of the "High school or less" category had significantly lower means than the other two groups (F[2,123] = 3.21; $p$ = 0.04).

## Factors' single effects: "Post-diagnosis drinking behavior" and "Drinking habits" most influential in acceptability judgements

A first round of results was obtained by considering factors 'single effects. To determine which factor was taken into account by participants, mean acceptability judgments were compared

**Table 2. Analysis of the two samples using factorial ANOVA performed with the indicated factor, either separately, or in 2-, 3- or 4-way interaction with other factors, as mentioned.**

| | Community ($N'$ = 132) | | Health pro. ($N''$ = 126) | |
|---|---|---|---|---|
| | $F$ | $\eta^2_p$ | $F$ | $\eta^2_p$ |
| Single factor | | | | |
| Drinking habits (H) | 114.99*** | 0.47 | 115.41*** | 0.48 |
| Diagnosis (D) | 3.12 | 0.02 | 17.48*** | 0.12 |
| Post-diagnosis behavior (B) | 349.75*** | 0.73 | 294.01*** | 0.70 |
| Physical activity (P) | 72.85*** | 0.36 | 39.59*** | 0.24 |
| Interactions between factors | | | | |
| H x D | 6.03* | 0.04 | 23.52*** | 0.16 |
| H x B | 58.84*** | 0.31 | 84.05*** | 0.40 |
| H x P | 6.16* | 0.04 | 7.54** | 0.06 |
| D x B | 24.98*** | 0.16 | 60.5*** | 0.33 |
| D x P | 2.86 | 0.02 | 0.31 | < 0.01 |
| B x P | 3.00 | 0.02 | 6.03* | 0.05 |
| H x D x B | 8.33** | 0.06 | 23.03*** | 0.16 |
| H x D x P | 0.81 | < 0.01 | 0.67 | < 0.01 |
| H x B x P | 4.21* | 0.03 | 3.77 | 0.03 |
| D x B x P | 0.09 | < 0.01 | 0.92 | < 0.01 |
| H x D x B x P | 0.01 | < 0.01 | 0.53 | < 0.01 |

* $p < .05$.

** $p < .01$

*** $p < .001$. η2p = partial eta squared.

using a within-subject factorial ANOVA (Table 2, "Single factor" part). The ANOVA was followed by post-hoc tests and the results were plotted, for improved readability (Fig 1).

Regarding the "Drinking habits" factor (Fig 1A), in the community sample, the character's behavior was judged more acceptable when she had one drink compared with 5–6 drinks ($M$ = 6.93 and 5.84, respectively; $p < 0.001$). In health professionals, Elisabeth's behavior was also considered more acceptable when she had one drink rather than 5–6 drinks ($M$ = 7.57 and 6.63, respectively; $p < 0.001$). In both samples, the effect size ($n^2$ p) was in the medium range (Table 2).

Considering the "Diagnosis" factor (Fig 1B), in the community sample, participants considered the character's behavior with similar acceptability levels when scenarios depicted "Polyps", "Early cancer" or "Advanced cancer" ($M$ = 6.37, 6.22 and 6.56, respectively; $p = 0.079$). In the health professional sample, participants considered less acceptable, with a small effect size, Elisabeth's behavior when the scenarios contained "Polyps" or "Early cancer" in comparison to "Advanced cancer" ($M$ = 6.97, 6.85 and 7.47, respectively; $p < 0.001$).

Regarding the "Post-diagnosis behavior" factor (Fig 1C), in community persons, the character's behavior was viewed as more acceptable when she "Quits drinking" compared with "Thinking about quitting drinking", and acceptability judgements were at their lowest level when she "Continues drinking" ($M$ = 8.39, 6.23 and 4.54, respectively; $p < 0.001$). Similar results were obtained in health professionals ($M$ = 8.93, 7.08 and 5.28, respectively; $p < 0.001$). This factor had a large effect size in both samples.

Concerning the "Physical activity" factor (Fig 1D), in the community sample, participants considered Elisabeth's behavior more acceptable when scenarios depicted the character as being physically active in comparison with a sedentary lifestyle ($M$ = 6.73 and 6.05,

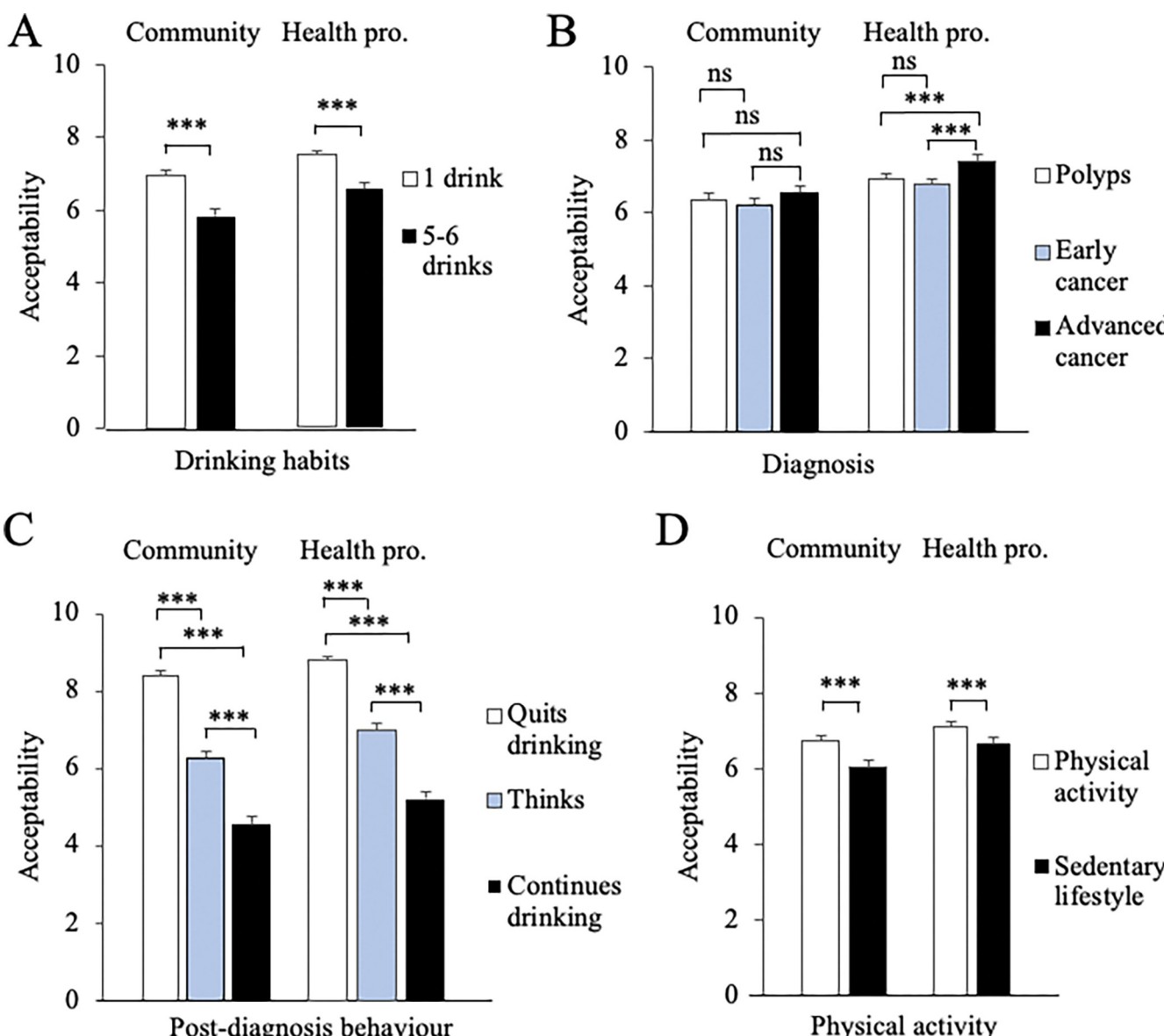

**Fig 1. A-D:** Acceptability judgments by community persons (left half of figures) and health professionals (right half) with respect to the factor indicated on the X axis (A: drinking habits, B: type of diagnosis, C: drinking behavior post-diagnosis, D: physical activity). The histograms represent mean ± standard error for each condition specified in the figure legend. Significance of differences between the indicated means was determined using factorial ANOVA and post-hoc: *, $p < .05$; **, $p < .01$; ***, $p < .001$; ns, not significant.

respectively; $p < 0.001$). Similar results were obtained in health professionals ($M = 7.34$ and 6.86, respectively; $p < 0.001$). This factor had a small effect size in both samples.

Altogether, these results showed that "Post-diagnosis drinking behavior" and "Drinking habits" were the most influential factors for judgment, in both samples.

## Factor interactions: "Continues drinking" is more unacceptable when associated with advanced cancer

In order to examine how factors interacted in participants' judgments, factorial ANOVAs of 2-, 3-, or 4-way interactions between factors were performed in each sample. The ANOVAs were

significant for a majority of 2-way interactions (Table 2, "Interactions between factors" part). In addition, a 3-way interaction (drinking Habits x Diagnosis x Behavior post-diagnosis, H x D x B) was significant in both samples. To further analyze this 3-way interaction, the means of interactions between these 3 factors were calculated, plotted then compared to each other using paired *t*-tests.

The main outcome of this analysis was that, in both community sample (Fig 2A) and health professional sample (Fig 2B), the condition "Continues drinking" was considered significantly more acceptable when the scenarios depicted a character with "Advanced cancer", when compared with "Polyps" or "Early cancer". This effect was observed whether the character had one or 5–6 drinks daily, even though the reversal was more pronounced in the "5–6 drinks" modality. The effect size was medium in the community sample (Cohen's *d* ranging from 0.34 to 0.36) and large in health professionals (Cohen's *d* between 0.78 and 0.82), with an acceptability rate almost doubled between "Early cancer" and "Advanced cancer" in the condition "Continues drinking 5–6 drinks". These results thus suggested that an advanced cancer stage, even when interacting with other factors, significantly restores the acceptability of drinking in health professionals, and, to a lower extent, in the general population.

## Discussion

The stigmatization around alcohol consumption might generate stigma towards colorectal cancer, representing an additional burden for patients. Reducing stigmatization around alcohol consumption may be thus important for colorectal cancer patients' treatments [28,30] and crucial in decreasing negative psychological effects [15,22,50]. However, identifying the factors involved in stigmatizing attitudes remains necessary. This report describes the first systematic study of factors influencing acceptability judgments regarding alcohol consumption in the context of colorectal cancer, based on a experimental method enabling the identification of variables involved in one's judgment [51,52]. This study was performed using two different samples, i.e., a community sample and a health professional sample.

### Respective influence of factors when analyzed separately

A first level of analysis of the data was achieved by considering single factor effects using a factorial ANOVA (Fig 1 and Table 2, "Single factor" part). This analysis was performed separately in each sample. These results showed that, in health professionals, each of the four factors studied (i.e., "Drinking habits", "Diagnosis", "Post-diagnosis behavior" and "Physical activity") was taken into account in participants' judgments, yet with important differences between factors. In the community sample, this analysis showed that only three factors ("Drinking habits", "Post-diagnosis behavior" and "Physical activity") were taken into account by participants when making their judgments, the "Diagnosis" factor not being taken into account.

In both samples, the most influential factors were "Post-diagnosis behavior" and "Drinking habits". Effectively, health professionals or community persons considered most acceptable the character's behavior when she was described as a person who quit drinking, while scenarios describing her as a person thinking about quitting drinking were much less acceptable, and those with a character who continued drinking had a very low acceptability rating. Similarly, scenarios depicting a character with the lowest alcohol consumption, that is, one drink per day, had higher acceptability levels, in comparison with those describing a character with high alcohol consumption. These results are in line with a previous study showing that heavy drinkers are more stigmatized than people who drink less [46]. In addition, our study provides a quantitative illustration concerning community persons' [45] and health professionals' [32] acceptability judgments surrounding alcohol consumption.

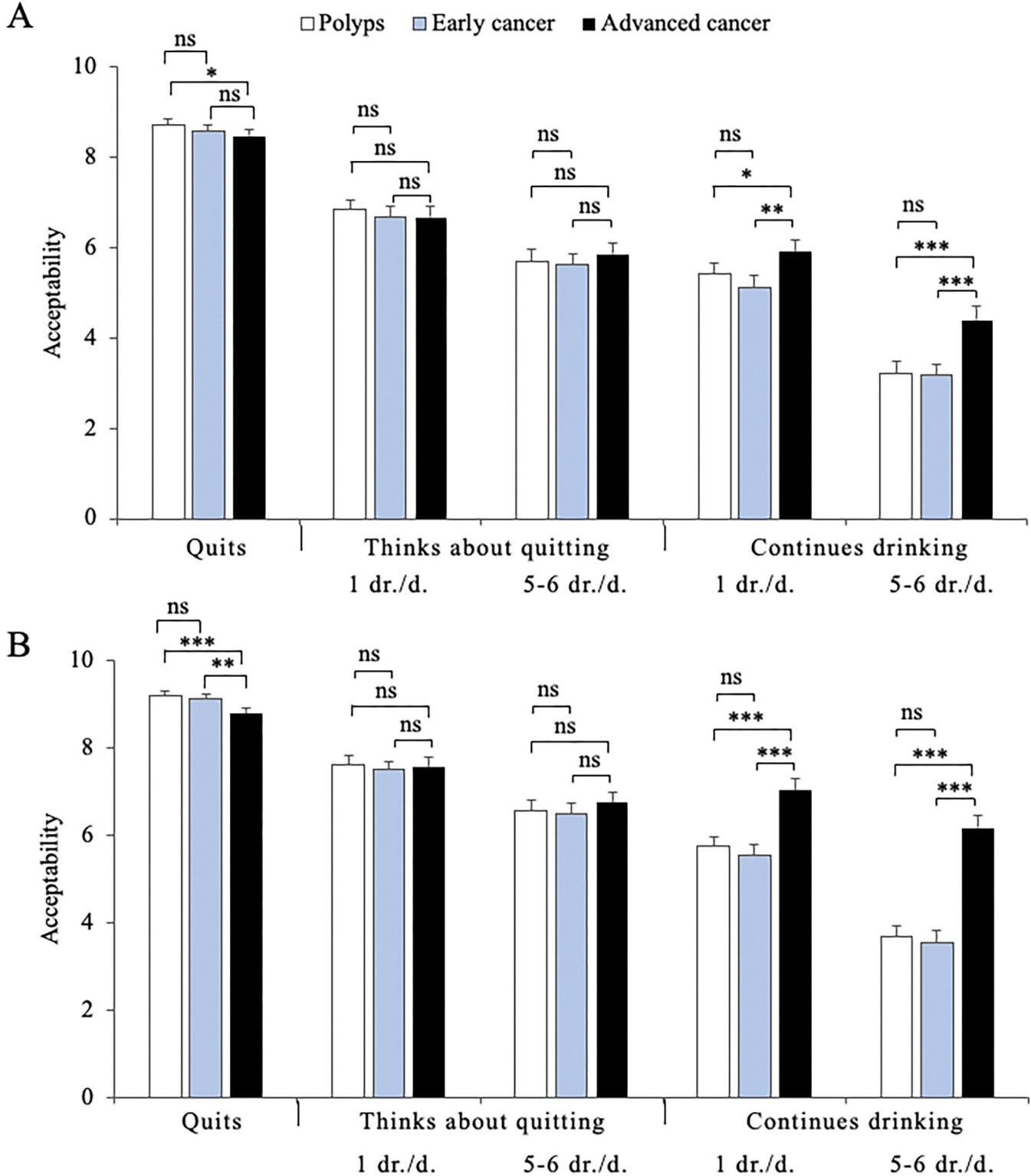

**Fig 2.** Acceptability judgments, by community persons (A) or health professionals (B), when combining the three following factors: Diagnosis (Polyps, Early cancer or Advanced cancer), Drinking habits (1 drink/day or 5–6 drinks/day) and Post-diagnosis behavior (Quits drinking, Thinks about quitting or Continues drinking). The histograms represent mean ± standard error for each condition specified in the figure legend. Significance of differences between the indicated means was determined using paired t-tests: *, $p < .05$; **, $p < .01$; ***, $p < .001$; ns, not significant.

Regarding the "Physical activity" factor, in both samples, participants considered more acceptable when scenarios described a character who engaged in physical activity than scenarios describing a sedentary character. This probably reflects the fact that physical activity has multiple benefits for patients having cancer [56] and that a sedentary lifestyle is a risk factor in the development of colorectal cancers [2]. This result could also be corroborated with a study showing that people made a link between physical activity and physical/mental well-being concerning colorectal cancer [61].

As to the "Diagnosis" factor, only health professionals seemed to have taken into account this factor in their judgments, as this sample found the character's behavior more acceptable when scenarios described her with a diagnosis of "Advanced cancer", in comparison with "Polyps" or "Early stage-cancer". In community sample, this factor had no influence on participants' judgments, except when it was considered in combination with other factors, as described below.

### Combined influence of factors

A second level of analysis was achieved by performing factorial ANOVAs of 2-, 3-, or 4-way interactions between factors (Fig 2 and Table 2, "Interaction between factors" part). Again, this analysis was performed separately in each sample. Interestingly, through this analysis, the "Diagnosis" factor was found, in both samples, to exert a significant effect when interacting with "Drinking habits" and "Post-diagnosis behavior". Indeed, when the scenarios depicted a character with "Advanced cancer", compared with "Early cancer" or "Polyps", "Continues drinking" was considered significantly more acceptable. This effect could be observed whether the character drank 1/5-6 drinks daily and was particularly prominent in health professionals. This suggested that the "Diagnosis/cancer stage" factor can significantly attenuate the poor acceptability of drinking alcohol in health professionals and, to a lower extent, in community persons. The reason for health professionals being more permissive than community persons remains to be identified but an hypothetical explanation could be that healthcare workers may be more aware than community persons of the fact that drinking cessation is most often a difficult challenge for a drinker [62]. Therefore, they probably consider that drinking cessation could represent an additional struggle for a character already facing an advanced cancer diagnosis.

### Limitations

This study's limitations include, firstly, the use of two moderate size convenience samples comprised of community persons or health professionals living in France. Generalization of the findings should thus be done with care. Secondly, the character described in the scenarios was always a female. Therefore, whether gender had an influence on participants' judgments was not explored, as testing the gender effect (e.g., woman *vs*. man) would have introduced an additional factor, requiring at least a doubling of the number of vignettes. Thirdly, the "drinking habits" factor could have been more precise: for instance, we could have included the beverage type (e.g., wine, beer or hard liquor), the precise amount of beverage and the alcohol percentage in the beverage.

### Clinical implications

Regarding clinical implications, this study illustrates with experimental data the acceptability judgments towards colorectal cancer patients when drinking subsists. In terms of implications for practice, these findings could allow to identify, as early as the time of diagnosis, the patients who are most at risk of being stigmatized, in order to reduce the impact of stigma. Indeed,

according to this study, the profile of the most stigmatized individuals would be those with early-stage colorectal cancer, who drank heavily before diagnosis, and who continue drinking after diagnosis.

## Conclusion

In conclusion, this report describes the first systematic study, in a community sample and a health professional sample, focusing on factors influencing acceptability judgments of drinking in the context of colorectal cancer. The most influential factor was the character's drinking behavior following her diagnosis. However, participants' attitudes towards drinking were more permissive, particularly in the health professional sample, when the character had an advanced rather than an early-stage cancer.

## Acknowledgments

The authors are very grateful to participants for their time and effort and to Anaëlle Preaubert for her help in data collection. CA, NC and PR designed the study and analyzed the data. CA collected the data. PR wrote the first version of the manuscript and NC and CA proofread it, making comments and changes. All authors approved the final version.

## Author Contributions

**Conceptualization:** Camille Auriol, Nicole Cantisano, Patrick Raynal.

**Data curation:** Camille Auriol, Patrick Raynal.

**Formal analysis:** Camille Auriol, Nicole Cantisano, Patrick Raynal.

**Funding acquisition:** Camille Auriol.

**Investigation:** Camille Auriol.

**Methodology:** Camille Auriol, Nicole Cantisano.

**Supervision:** Nicole Cantisano, Patrick Raynal.

**Writing – original draft:** Patrick Raynal.

**Writing – review & editing:** Camille Auriol, Nicole Cantisano.

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
