## [Decision Letter · Decision Letter 0]

20 Nov 2023

PONE-D-23-13716Factors influencing the acceptability of drinking for a patient with colorectal cancerPLOS ONE

Dear Dr. Auriol,

Thank you for submitting your manuscript to PLOS ONE. After careful consideration, we feel that it has merit but does not fully meet PLOS ONE’s publication criteria as it currently stands. Therefore, we invite you to submit a revised version of the manuscript that addresses the points raised during the review process.

We look forward to receiving your revised manuscript.

Kind regards,

Gunasekara Vidana Mestrige Chamath Fernando,

MBBS PgD-FM DipPallMed MCGP MRCGP

Academic Editor

PLOS ONE

Journal Requirements:

"CA was supported by a fellowship program from Institut National du Cancer (INCa_15747 SPADOC20-02).

https://www.e-cancer.fr/"          

Additional Editor Comments:

Dear Camille Auriol,

Thank you for submitting the manuscript for the PLOS One journal's consideration. While the area researched is extremely important, I wish to inform you that the two independent reviewers have had starkly contrasting views regarding your submission, with one reviewer rejecting the manuscript.

However, I have curated and modified their comments and am listing them below. Please make necessary amendments in the manuscript with track changes and signpost to these changes in a table of rectifications in a separate document. Where changes cannot be made, I request you make ample justifications for not doing so. In addition, you are expected to submit a clean version of the manuscript.

Reviewer 1

1. When presenting the rationale of the study the authors describe possible consequences of stigma on persons living with colorectal cancer. However, as a reader, I would also like to know the beliefs and thoughts underlying stigma attitudes towards those patients and how is this stigma expressed.

2. The way the authors presented the theoretical framework is misleading: Functional Theory of Cognition is not a “method enabling to identify variables involved in one’s judgment”, but rather a theory of how individuals process information to form judgments.

Reference : Anderson NH. A functional theory of cognition. Psychology Press; 2014.

3. Page 4 Line 100-102 : “The scenarios were based on four "within subject factors" supposed to have an impact on the perception of a person who drinks alcoholic beverages and who is diagnosed with colorectal cancer”.

Vignette-based methods require that factor selection for the scenarios should be based on evidence, not supposition. No previous research suggested that factors selected by the authors--“Drinking habits”, “Drinking behavior”, “Type of cancer diagnosis and prognosis” and level of “Physical activity” -- influence attitudes towards cancer patients with alcohol consumption habits.

4. The authors should provide information regarding recruitment of the participants: Where was the sample of health professionals recruited? Where was the sample of lay people recruited? Where were the participants approached? What was the recruitment procedure for each sample?

5. The research material is unclear. For instance, the understanding of the expression “drinks of alcohol beverages” in the vignette is confusing. Assessment of alcohol consumption cannot be based on the number of drinks solely. One important factor is the alcohol level in the beverage. 3 glasses of beer do not equal 3 glasses of whisky when assessing alcohol consumption.

6. The authors developed the rationale of their study and the discussion of their findings around stigma. However, their research did not investigate the issue of stigma. Their research question has focused on acceptability. Stigma and acceptability are two very different concepts.

Reviewer 2

Please find the comments below:

1. I would highly recommend the authors to include 'alcohol drinking' instead of 'drinking' in the title to increase the visibility.

2. The authors have taken two type population - community persons and health professionals. I am very interested to know if there is any difference in the acceptability judgment between highly educated and low/no educated person among these population. (Editor: please clarify or add to the present paper)

3. Did the authors find any difference in the acceptability judgment between male and female study participants? (Editor: please clarify or add to the present paper)

Reviewers' comments:

Reviewer's Responses to Questions

**Comments to the Author**

1. Is the manuscript technically sound, and do the data support the conclusions?

Reviewer #1: Yes

Reviewer #2: No

2. Has the statistical analysis been performed appropriately and rigorously? 

Reviewer #1: Yes

Reviewer #2: I Don't Know

3. Have the authors made all data underlying the findings in their manuscript fully available?

Reviewer #1: Yes

Reviewer #2: Yes

4. Is the manuscript presented in an intelligible fashion and written in standard English?

Reviewer #1: Yes

Reviewer #2: No

---

## [Author Response · Author response to Decision Letter 0]

8 Dec 2023

Response to Reviewers

We thank very much the Academic Editor and the Reviewers for their relevant and helpful comments, which we believe have improved our manuscript’s quality. Please find below a document that responds to each point raised by the Academic Editor and Reviewers. 

Reviewer 1

1. When presenting the rationale of the study the authors describe possible consequences of stigma on persons living with colorectal cancer. However, as a reader, I would also like to know the beliefs and thoughts underlying stigma attitudes towards those patients and how is this stigma expressed.

Response: We thank Reviewer 1 for this relevant comment that helped us improving the manuscript. We thus have added three sentences and references on beliefs and thoughts that underlie stigmatizing attitudes towards patients with colorectal cancer (lines 74 to 79).

2. The way the authors presented the theoretical framework is misleading: Functional Theory of Cognition is not a “method enabling to identify variables involved in one’s judgment”, but rather a theory of how individuals process information to form judgments.

Reference : Anderson NH. A functional theory of cognition. Psychology Press; 2014.

Response: We thank Reviewer 1 for this clarification on the Functional Theory of Cognition. This part was modified accordingly (lines 106 to 108).

3. Page 4 Line 100-102 : “The scenarios were based on four "within subject factors" supposed to have an impact on the perception of a person who drinks alcoholic beverages and who is diagnosed with colorectal cancer”.

Vignette-based methods require that factor selection for the scenarios should be based on evidence, not supposition. No previous research suggested that factors selected by the authors--“Drinking habits”, “Drinking behavior”, “Type of cancer diagnosis and prognosis” and level of “Physical activity” -- influence attitudes towards cancer patients with alcohol consumption habits.

Response: We are grateful to Reviewer 1 for this valuable comment. In this manuscript’s new version, we have developed the rationale regarding the choice of factors for the construction of the scenarios (lines 119 to 122 and 126 to 135).

4. The authors should provide information regarding recruitment of the participants: Where was the sample of health professionals recruited? Where was the sample of lay people recruited? Where were the participants approached? What was the recruitment procedure for each sample?

Response: We have added information regarding the recruitment of participants, in the Participants section (lines 146 to 149).

5. The research material is unclear. For instance, the understanding of the expression “drinks of alcohol beverages” in the vignette is confusing. Assessment of alcohol consumption cannot be based on the number of drinks solely. One important factor is the alcohol level in the beverage. 3 glasses of beer do not equal 3 glasses of whisky when assessing alcohol consumption.

Response: We agree with this comment, even though it would have been difficult to take into account the type of alcoholic beverage, the volume taken and the percentage of alcohol in the drink. We now mention this aspect as a limitation (lines 376 to 378).

6. The authors developed the rationale of their study and the discussion of their findings around stigma. However, their research did not investigate the issue of stigma. Their research question has focused on acceptability. Stigma and acceptability are two very different concepts.

Response : We are grateful to Reviewer 1 for this comment. Indeed, stigma and acceptability are different concepts. To clarify this, we have further developed the definition of stigma in the Introduction section and clarified the relationship between stigma and acceptability (line 68 to 72). In addition, for improved accuracy of the text, the term “stigmatization” was changed to “acceptability judgments” in the results section.

Reviewer 2

Please find the comments below:

1. I would highly recommend the authors to include 'alcohol drinking' instead of 'drinking' in the title to increase the visibility.

Response: We thank Reviewer 2 for this valuable suggestion. The title and short title were modified accordingly.

2. The authors have taken two type population - community persons and health professionals. I am very interested to know if there is any difference in the acceptability judgment between highly educated and low/no educated person among these population. (Editor: please clarify or add to the present paper)

Response: We thank Reviewer 2 for this suggestion. We have now included the results regarding the influence of education level on acceptability judgments in each sample. These results are presented within the last paragraph of the sub-section entitled "Participants’ characteristics and descriptive statistics" in the Results section (lines 232 to 241).

3. Did the authors find any difference in the acceptability judgment between male and female study participants? (Editor: please clarify or add to the present paper)

Response: We apologize for this oversight. We now include t-test results regarding the lack of statistical significance of differences of acceptability judgements between male and female (lines 228 and 231).

---

## [Editor Report · Decision Letter 1]

13 Dec 2023

Factors influencing the acceptability of alcohol drinking for a patient with colorectal cancer

PONE-D-23-13716R1

Dear Dr. Auriol,

We’re pleased to inform you that your manuscript has been judged scientifically suitable for publication and will be formally accepted for publication once it meets all outstanding technical requirements.

Kind regards,

Gunasekara Vidana Mestrige Chamath Fernando, MBBS PgD-FM DipPallMed MCGP MRCGP

Academic Editor

PLOS ONE

Additional Editor Comments (optional): The first reviewer's fifth comment remains an unmodifiable factor favouring rejection as proposed. However, considering that the participants' subjective perception about the level of alcohol consumption is still a valid measurement, I believe this paper deserves publication. 
---

## [Editor Report · Acceptance letter]

18 Dec 2023

PONE-D-23-13716R1 

PLOS ONE

Dear Dr. Auriol, 

I'm pleased to inform you that your manuscript has been deemed suitable for publication in PLOS ONE. Congratulations! Your manuscript is now being handed over to our production team.

Kind regards, 

on behalf of

Dr Gunasekara Vidana Mestrige Chamath Fernando 

Academic Editor

PLOS ONE